# The First Find of $Cr_2O_3$ Eskolaite Associated with Marble-Hosted Ruby in the Southern Urals and the Problem of Al and Cr Sources

**Aleksander Kissin [1,2,*], Irina Gottman [1,2], Sergei Sustavov [2], Valery Murzin [1] and Daria Kiseleva [1]**

[1] A.N. Zavaritsky Institute of Geology and Geochemistry, Ural Branch of Russian Academy of Sciences, Vonsovskogo str., 15, 620016 Ekaterinburg, Russia; gottman@igg.uran.ru (I.G.); murzin@igg.uran.ru (V.M.); Kiseleva@igg.uran.ru (D.K.)

[2] Department of Mineralogy, Petrology and Geochemistry, Ural State Mining University, Kuybysheva str.,30, 620144 Ekaterinburg, Russia; sustavov.s@ursmu.ru

\* Correspondence: kissin@igg.uran.ru

**Abstract:** The results of the study of eskolaite associated with marble-hosted ruby found for the first time in the Kuchinskoe occurrence (Southern Urals) are presented. Here, eskolaite was located on the surface and near-surface regions of ruby crystals. Eskolaite diagnostics was confirmed by powder X-ray diffraction (URS-55). The morphology and chemical composition of eskolaite and associated ruby was studied using a JSM-6390LV scanning electron microscope and a Cameca SX 100 electron probe microanalyzer. The eskolaite crystals were hexagonal and tabular, up to 0.2 mm in size. Ruby mineralization was formed during prograde and retrograde dynamothermal metamorphism. The eskolaite associated with the prograde stage ruby contained $Al_2O_3$ (9.1–23.62 wt %), $TiO_2$ (0.52–9.66 wt %), $V_2O_3$ (0.53–1.54 wt %), FeO (0.03–0.1 wt %), MgO (0.05–0.24 wt %), and $SiO_2$ (0.1–0.21 wt %). The eskolaite associated with the retrograde stage ruby was distinguished by a sharp depletion in Ti and contained $Al_2O_3$ (12.25–21.2 wt %), $TiO_2$ (0.01–0.07 wt %), $V_2O_3$ (0.32–1.62 wt %), FeO (0.01–0.08 wt %), MgO (0.0–0.48 wt %), and $SiO_2$ (0.01–0.1 wt %). The associated rubies contained almost equal amounts of $Cr_2O_3$ (2.36–2.69 wt %) and were almost free from admixtures. The identification of the eskolaite associated with the marble-hosted rubies from the Kuchinskoe occurrence is a new argument in favor of introduction of Al and Cr into the mineral formation zone. The mineralization was localized in the metamorphic frame of the granite gneiss domes and was formed synchronously with them.

**Keywords:** gems; ruby; marble; eskolaite; Southern Urals

## 1. Introduction

The issue of Al and Cr sources in ruby (a corundum containing a chromium isomorphic impurity) deposits in calcite (dolomite) marbles is actively discussed in the scientific literature. Various ideas have been studied on this issue: (1) the lateritic weathering of an impure limestone [1–4]; (2) the lenses of bauxite-like sediments in limestones [5]; (3) the desilication of primary sedimentary rocks during regional alkali metasomatism [6]; (4) the introduction of Al and Cr by metamorphogenic fluids during rock granitization and dynamothermal metamorphism [7,8]; (5) the introduction of Al using the gas phase of deep fluids during alkaline magmatism [9]; (6) the introduction of Al as a result of decompression during deep tectonic erosion [10]; and (7) the redistribution of Al and Cr during the metamorphism of sedimentary limestones with evaporitic lenses [11,12]. A review of the views on this issue was given by Giluani et al. [13]. All researchers noted the presence of Cr-containing minerals

in ruby-bearing marble, such as muscovite (fuchsite), phlogopite, margarite, pargasite, diopside, tourmaline, diaspore, rutile, sphene, and red spinel. In the Middle Ural Mountains (Urals, Russia), uvarovite with a grossular admixture ($Cr_2O_3$ up to 18.60 wt %) is found in ruby-bearing marbles [14]. The article reports on the discovery of chromium oxide in the intergrowth with ruby, which can be used in discussions about the sources of Al and Cr in ruby-bearing marbles.

Natural chromium oxide ($Cr_2O_3$) of supposedly hydrothermal-metasomatic genesis was discovered in the middle of the last century in the Outokumpu Cu-Co-Zn skarn deposit in Finland and was called eskolaite [15]. Hydrothermal-metasomatic eskolaite was also described in the chromitites of the Mariinsky and Bazhenovsky ophiolite massifs in the Middle Urals. In the Mariinsky emerald deposit, eskolaite formed inclusions in mariinskite (a chromium analogue of chrysoberyl) associated with fluorophlogopite, Cr-containing muscovite, and tourmaline [16]. In the Bazhenovsky massif, eskolaite was found in alumina chromitites [17] as well as in the form of inclusions in mariinskite [18].

The eskolaite of magmatic origin was found in the chromite ores of the Rai-Iz and Voikar–Syninsky hyperbasite massifs in the Polar Urals [19]. It formed as a result of fractionation of sulfide and chromospinelide ores from a silicate melt at temperatures 900–1100 °C The magmatic eskolaite was also described in magnetite-ilmenite ores of the Kusa gabbro intrusion (Southern Urals) [20].

Eskolaite was found in kimberlites [21–23]. The eskolaite intergrown with diamond and with picrochromite inclusions was reported for the Udachnaya kimberlite pipe (Yakutia) [24].

In the Cis-Baikal region, the eskolaite and karelianite ($V_2O_3$) of metamorphic genesis were revealed in the thin layers of graphite-enriched sillimanite-cordierite quartzite schists of the Olkhon series [25]. The grain size was 5–20 μm in intergrowth with rutile, $(Cr,V)_2Ti_3O_9$ olkhonskite, and $V_2Ti_3O_9$ schreyerite. In the metacarbonate rocks of the Slyudyanka series (Southern Cis-Baikal), the eskolaite was described in calcite-quartz-diopside rocks [26]. The eskolaite was accompanied by magnesiochromite and $(Cr,V)_2O_3$ karelianite-eskolaite. The authors of [26] believe that Cr and V were introduced into paleo-sediments by chemical deposition. Then, during prograde granulite facies metamorphism, Cr and V participated in the formation of magnesiochromite, eskolaite, and karelianite. The formation of eskolaite in metamorphic rocks is possible even at low chromium concentrations in the substrate, but under the conditions of isochemical metamorphism, Cr inertness, and the absence of its isomorphic incorporation into co-crystallizing minerals.

This article presents the results of the studies of eskolaite from the Kuchinskoe marble-hosted ruby occurrence in the Southern Urals, a mineral phase that has not been previously described in association with marble-hosted ruby.

## 2. Geological Setting

The Kuchinskoe marble-hosted ruby occurrence is located in the Kochkar anticlinorium (Southern Urals) (Figure 1). The anticlinorium runs in the submeridional direction for 140 km, with a width up to 28 km [27]. The tectonic boundaries are represented by thrusts dipping under the adjacent synclinorium structures. The anticlinorium is characterized by the presence of granite gneiss domes, which form the centres of dynamothermal metamorphism [28–30]. Crystalline schists, amphibolites, and marbles intruded by the dikes of granites and pegmatites occur in the metamorphic frame of the domes [31]. The metamorphism is syntectonic: prograde in the Carboniferous period and retrograde in the Permian-Triassic period. The level of metamorphism in the dome structures reached the conditions of the amphibolite facies and in the inter-dome structures reached the epidote-amphibolite facies (according to mineral paragenesis) [28,29].

The geology of the Kuchinskoe marble-hosted ruby occurrence was studied by one of the authors (A.K.) from 1979 to 1988 as a field geologist performing geological explorations. The ruby-bearing marbles of the Kuchinskoe occurrence are localized in the inter-dome structure. The marbles are banded, cleaved, and repeatedly recrystallized [32]. At the deposit: (1) coarse-grained calcite marble is the most prevalent, while (2) Mg-calcite marble (which can be schisted) is found locally, among which the bodies of (3) dolomite-calcite/calcite-dolomite marble are found. The marbles are split by dikes of

granite and pegmatite. Type 1 and 2 marbles existed at the time of the dike introduction. The dike was not tectonically disturbed in the time intervals of its contact with these marbles. Consequently, the cleaving and schisting of these marbles took place before the dike introduction. There are no quenching zones in the dike and no marble recrystallization at the contact. This is explained by the similar temperatures of the embedded granite solution-melt and the host rocks. At the contacts with the massive two-carbonate marble, dikes are disturbed by low-amplitude shears and a zone of forsterite skarn 1.5–3.0 cm wide is formed along the marble. Consequently, two-carbonate marble was formed after the introduction of the dike. The dikes are genetically related to anatectic granites in the apical part of the granite gneiss domes. The time of their massive introduction records the time of stress relief and the drop in all-round pressure. Thus, 1 and 2 type marbles were formed during prograde metamorphism, while type 3 marble was formed during retrograde metamorphism [7,8,33,34].

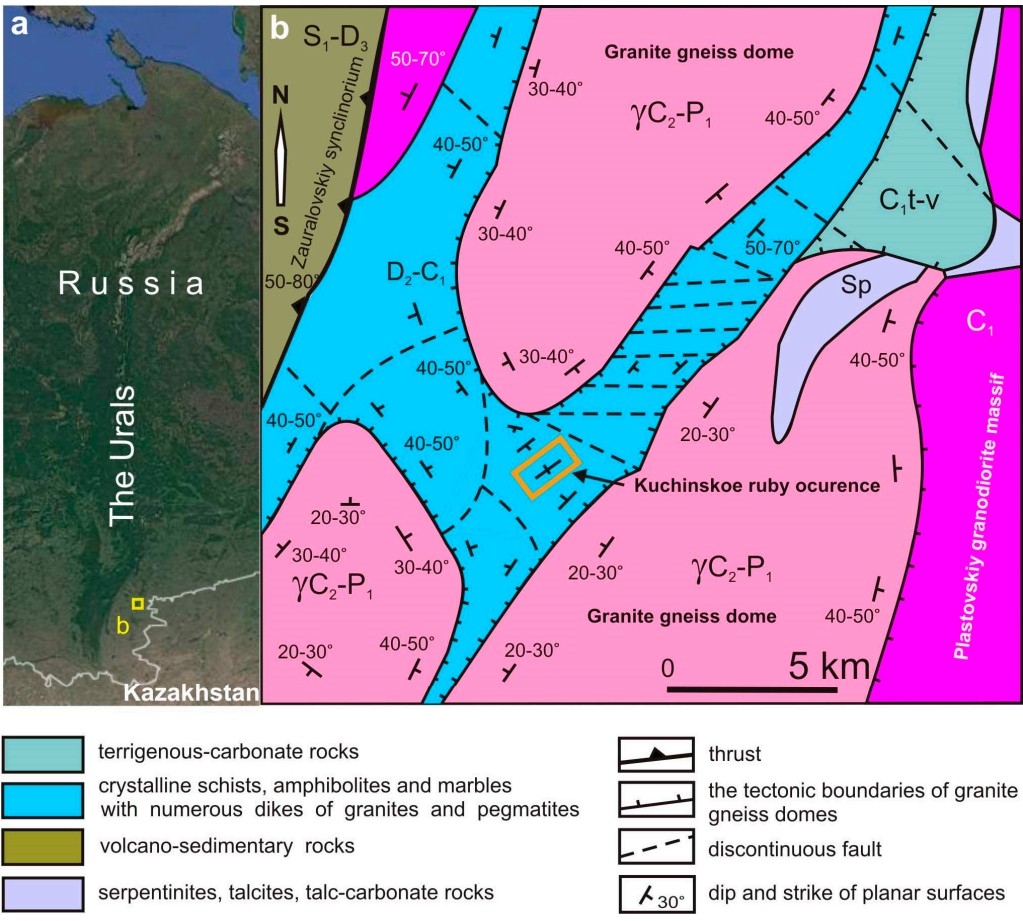

**Figure 1.** (**a**) The localization of the Kuchinskoe marble-hosted ruby occurrence and (**b**) a schematic geological map of the site.

Type 1 rubies are dark red crystals of tabular shape with the highest $Cr_2O_3$ content (1.22–2.91 wt %). They are found in the form of disseminated impregnation in Mg-calcite marble and are replaced by spinel in dolomite marble. They are accompanied by pyrite, anhydrite, and apatite. Type 2 rubies are pink, short-columnar, rounded crystals with the lowest $Cr_2O_3$ content (0.11–0.51 wt %). They are found in the form of disseminated impregnation in dolomite-calcite/calcite-dolomite marble and accompanied by rutile and pyrite, often replaced with pink spinel associated with graphite, forsterite, and norbergite. Type 3 rubies (more correctly called corundum) are represented by long-prismatic crystals and irregular-shaped grains of red, blue, violet, and white color, and are often polychromatic; the red corundum is close to type 1 rubies in terms of $Cr_2O_3$ contents (0.03–2.97 wt %). These corundums are

confined to cleavage cracks in marbles and are accompanied by colorless phlogopite, pyrite, pyrrhotite, sphalerite, and other minerals. They are replaced by spinel associated with fluorite [7,8].

Karst formations filled with Paleogene–Neogene kaolin-montmorillonite clays mixed with quartz sand are widely developed in the area of marble distribution in the Kochkar anticlinorium. All formations were studied by drilling and tested during ruby prospecting. In karst sediments, ruby and related minerals (fuchsite, phlogopite, Cr-pargasite, red spinel, and others) are found only if ruby-bearing marbles are present under the karst. There are no signs of abrasion on the surfaces of ruby crystals and associated minerals, which indicates the absence of their transportation by water flows.

## 3. Materials and Methods

The authors had at their disposal several kilograms of crystals and the grains of red corundum (ruby and pink sapphire) of non-jewellery quality, 2–20 mm in size. The investigated rubies were extracted from karst formations in marbles. Corundum crystals from the Kuchinskoe marble quarries did not exceed 9 mm in size. While the largest crystal from karst formations reaches 5 cm, crystals up to 4 mm prevail. The types of corundum (see geological setting) from karst are easily diagnosed and correspond to the types of corundum from the underlying marbles. Their colors, transparency, inclusions, chemical composition, as well as the mineralogy of the intergrowths were also identical. Sometimes corundum mineralization of marbles can also be traced in the karst clays lying on the surface of the mines. In the surroundings of the Kuchinskoe occurrence, other primary sources of corundum are not known. The studied rubies are genetically associated with marbles, as is evidenced by the studies of oxygen isotopes [35].

Ruby grains with the largest aggregates of an unknown mineral were studied by scanning electron microscopy (SEM) and electron probe microanalysis (EPMA) at the Geoanalitik Collective Use Center of the Institute of Geology and Geochemistry of the Ural Branch of the Russian Academy of Sciences (Ekaterinburg). The X-ray powder diffraction analysis of the eskolaite structure was performed at the Ural State Mining University (Ekaterinburg).

Powder X-ray diffraction studies were performed on an URS-55 X-Ray powder diffraction spectrometer using the following operational conditions: RKD chamber, $Fe_{K\alpha+\beta}$ radiation, voltage 30 kV, current 10 mA. Eskolaite powder was mixed with rubber glue. A cylinder was made from this mixture, which was fixed in the sample holder. The analysis was carried out using a photo method. The results are shown in Table 1.

The chemical composition of the minerals was studied on a CAMECA SX 100 electron probe microanalyzer (Cameca, Gennevilliers, France). The analyses were carried out with an accelerating voltage of 15 kV, a current of 10 nA for Al, 20 nA for Cr, and 150 nA for Fe, Ti, Mg, Si, and V, with an electron beam diameter of 1–3 μm on the sample. The following standard samples were used: $Cr_2O_3$, $Al_2O_3$, $Fe_2O_3$, $MgO$, $TiO_2$, $V_2O_5$, and $SiO_2$. To identify all the peaks, the most intense Kα lines were used. The superposition of the V Kα line on Ti Kβ was taken into account. Al, Mg, Ga were measured using TAP crystals, chromium–LPET, titanium and vanadium-PET, and iron-LLIF crystal. The duration of the intensity measurement at the peaks of the analytical lines and the background on each side of the peak were: for the major elements—Al and Cr—10 and 5 s, respectively; for Fe, Ti, Mg, Si, Al, Cr 30 and 15 s; and for V 60 and 30 s. Standard deviation of the elemental content (wt % ) was less than 0.5 for Cr, less than 0.6 for Al, less than 0.01 for Fe, Ti, Mg, Si, and V, 0.02 for Ga, less than 0.06 for Al (in eskolaite), and 0.05 for Cr (in corundum). Three polished sections of ruby with eskolaite were studied (Table 2). Sample 1 was a type 1 ruby (analyses 1 and 2) with 3 inclusions (analyses 3–5). Sample 2 was a type 3 ruby (analyses 6–9) with 6 inclusions (analyses 10–15). Sample 3 was a type 3 ruby (analyses 16–18) with 5 inclusions (analyses 19–23).

**Table 1.** Eskolaite X-Ray diffraction data.

| Eskolaite JCPDS | | | Eskolaite with Ruby | |
|---|---|---|---|---|
| *d (Å)* | *I/I$_1$* | *hkl* | *I* | *d$_{\alpha/n}$* |
| 3.633 | 74 | 012 | 6 | 3.64 |
| 2.666 | 100 | 104 | 10 | 2.67 |
| 2.480 | 96 | 110 | 10 | 2.48 |
| 2.264 | 12 | 006 | 1 | 2.26 |
| 2.176 | 38 | 113 | 3 | 2.16 |
| 2.048 | 9 | 202 | - | - |
| 1.816 | 39 | 024 | 4 | 1.811 |
| 1.672 | 90 | 116 | 10 | 1.663 * |
| 1.579 | 13 | 122 | 2 | 1.565 |
| 1.465 | 25 | 214 | 3 | 1.462 |
| 1.4314 | 60 | 300 | 7 | 1.426 * |
| 1.2961 | 20 | 1.0.10 | 1 | 1.286 |
| 1.2398 | 17 | 220 | 1 | 1.233 |
| 1.2101 | 7 | 306 | - | - |
| 1.1731 | 14 | 128, 312 | - | - |
| 1.1488 | 10 | 0.210 | 3 | 1.144 |
| 1.1239 | 10 | 134 | - | - |
| 1.0874 | 17 | 226 | 10 | 1.083 |
| 1.0422 | 16 | 2.1.10 | 8 | 1.038 |

X-Ray diffraction pattern contains 9 lines not included in the Table

| a$_o$ = 4.954 | a$_o$ = 4.94 |
|---|---|
| c$_o$ = 13.584 | c$_o$ = 13.51 |

* Reflections used for unit cell parameters calculation.

**Table 2.** The chemical composition of eskolaite and associated ruby of the prograde and retrograde metamorphism according to X-ray microanalysis.

| No | Object | Oxide (wt %) | | | | | | | | Crystal Chemical |
|---|---|---|---|---|---|---|---|---|---|---|
| | | TiO$_2$ | FeO | V$_2$O$_3$ | MgO | Al$_2$O$_3$ | SiO$_2$ | Cr$_2$O$_3$ | Total | Formula |
| Prograde Metamorphism | | | | | | | | | | |
| 1 | Ruby 1 | bdl | bdl | bdl | bdl | 95.95 | 0.07 | 2.91 | 98.93 | Cr$_{0.04}$Al$_{1.96}$O$_3$ |
| 2 | Ruby 1 | bdl | bdl | bdl | bdl | 97.51 | bdl | 2.47 | 99.98 | Cr$_{0.03}$Al$_{1.97}$O$_3$ |
| 3 | Esk | 9.66 | bdl | 1.54 | 0.24 | 9.1 | 0.1 | 78.21 | 98.9 | Cr$_{1.52}$Al$_{0.26}$V$_{0.03}$Ti$_{0.18}$Mg$_{0.01}$O$_3$ |
| 4 | Esk | 0.52 | | 0.53 | 0.15 | 23.62 | 0.14 | 73.3 | 98.37 | Cr$_{1.34}$Al$_{0.64}$V$_{0.01}$Mg$_{0.01}$O$_3$ |
| Retrograde Metamorphism | | | | | | | | | | |
| 5 | Ruby 3 | bdl | bdl | 0.04 | bdl | 97.12 | 0.04 | 2.16 | 99.36 | Cr$_{0.03}$Al$_{1.97}$O$_3$ |
| 6 | Ruby 3 | bdl | bdl | 0.11 | bdl | 97.39 | 0.07 | 1.78 | 99.35 | Cr$_{0.02}$Al$_{1.98}$O$_3$ |
| 7 | Ruby 3 | bdl | bdl | bdl | bdl | 96.2 | 0.03 | 2.97 | 99.20 | Cr$_{0.04}$Al$_{1.96}$O$_3$ |
| 8 | Ruby 3 | bdl | bdl | bdl | bdl | 95.79 | 0.03 | 2.67 | 98.49 | Cr$_{0.04}$Al$_{1.96}$O$_3$ |
| 9 | Esk | bdl | 0.08 | 1.37 | 0.16 | 12.25 | bdl | 83.76 | 97.64 | Cr$_{1.62}$Al$_{0.35}$V$_{0.03}$O$_3$ |
| 10 | Esk | bdl | 0.08 | 0.32 | 0.08 | 16.15 | 0.1 | 82.08 | 98.82 | Cr$_{1.54}$Al$_{0.45}$V$_{0.01}$O$_3$ |
| 11 | Esk | bdl | bdl | 0.62 | bdl | 17.81 | bdl | 80.34 | 98.8 | Cr$_{1.49}$Al$_{0.49}$V$_{0.02}$O$_3$ |
| 12 | Esk | bdl | bdl | 0.56 | 0.08 | 18.5 | bdl | 80.23 | 99.41 | Cr$_{1.48}$Al$_{0.51}$V$_{0.01}$O$_3$ |
| 13 | Esk | 0.07 | bdl | 1.62 | bdl | 19.11 | bdl | 78.69 | 99.52 | Cr$_{1.45}$Al$_{0.52}$V$_{0.03}$O$_3$ |
| 14 | Esk | bdl | 0.06 | 0.93 | 0.12 | 21.2 | bdl | 77.6 | 99.96 | Cr$_{1.41}$Al$_{0.57}$V$_{0.02}$O$_3$ |
| 15 | Ruby 3 | bdl | bdl | bdl | bdl | 97.13 | 0.03 | 2.64 | 99.80 | Cr$_{0.04}$Al$_{1.96}$O$_3$ |
| 16 | Ruby 3 | bdl | bdl | bdl | bdl | 97.48 | 0.07 | 2.40 | 99.95 | Cr$_{0.03}$Al$_{1.97}$O$_3$ |
| 17 | Ruby 3 | bdl | bdl | bdl | bdl | 96.97 | 0.03 | 2.40 | 99.40 | Cr$_{0.03}$Al$_{1.97}$O$_3$ |
| 18 | Esk | 0.02 | 0.09 | 0.73 | 0.17 | 18.0 | bdl | 79.4 | 98.41 | Cr$_{1.48}$Al$_{0.50}$V$_{0.01}$Mg$_{0.01}$O$_3$ |
| 19 | Esk | | 0.02 | 0.39 | 0.12 | 26.02 | bdl | 73.74 | 100.3 | Cr$_{1.30}$Al$_{0.69}$V$_{0.01}$O$_3$ |
| 20 | Esk | bdl | 0.02 | 1.5 | 0.14 | 24.59 | bdl | 72.95 | 99.2 | Cr$_{1.31}$Al$_{0.66}$V$_{0.03}$O$_3$ |
| 21 | Esk | bdl | 0.02 | 1.94 | 0.26 | 13.39 | bdl | 82.08 | 97.69 | Cr$_{1.57}$Al$_{0.38}$V$_{0.04}$Mg$_{0.01}$O$_3$ |
| 22 | Esk | | bdl | 0.6 | 0.21 | 25.61 | bdl | 73.75 | 100.18 | Cr$_{1.30}$Al$_{0.68}$V$_{0.01}$Mg$_{0.01}$O$_3$ |

bdl: the values below the detection limit.

The sections of ruby grains with eskolaite as well as ruby grain surface morphology were studied using scanning electron microscopy (SEM). Back scattered electron images were obtained by a JSM-6390LV electronic scanning microscope (JEOL, Akishima, Tokyo, Japan) with an INCA Energy 450 X-Max 80 energy dispersive spectrometer (Oxford Instruments, Abingdon, Great Britain). Operating conditions were an acceleration voltage of 20 kV and an exposure time of 5 ms per pixel.

## 4. Results

SEM indicated that the dark colored crusts on a ruby were represented by $Cr_2O_3$ eskolaite with Al, Fe, Ti, and V admixtures. The eskolaite was represented by an aggregate of small tabular crystals (Figures 2–4) up to 10 microns in size, rarely up to 0.2 mm. More often, crystals of diaspore, spinel, rutile, and green tourmaline were observed on the surface of ruby crystals. The growth of eskolaite crystals on corundum is chaotic and tends to develop along and within surface imperfections on the ruby crystal faces and on their near-surface part. X-ray diffraction studies confirmed the diagnosis of the mineral (Table 1). The results of the X-ray microanalysis of eskolaite and associated ruby are given in Table 2. The admixtures of Al, Ti, V, and sometimes Fe, widely varying in content, are common for eskolaite.

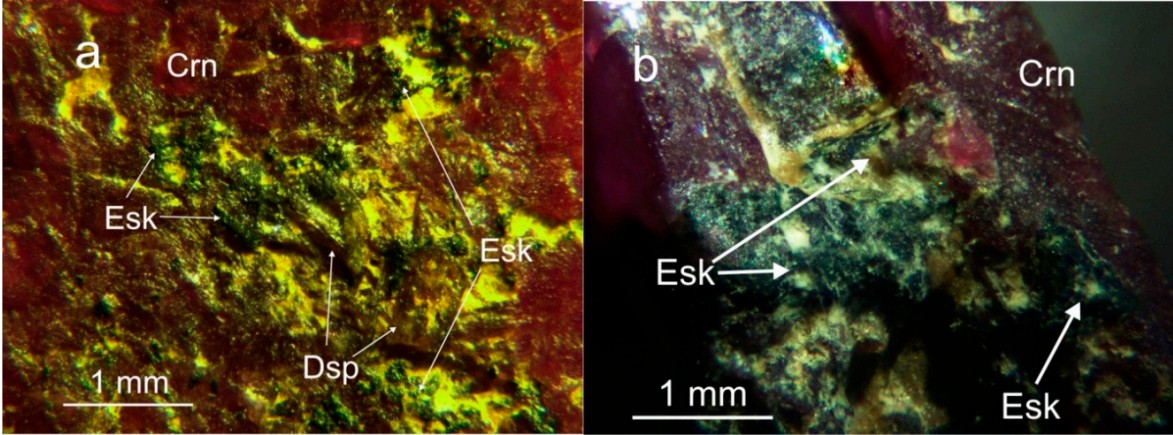

**Figure 2.** Eskolaite (Esk) and diaspore (Dsp) on the (**a**) pinacoidal and (**b**) rhombohedral crystal faces of the prograde stage ruby (Crn).

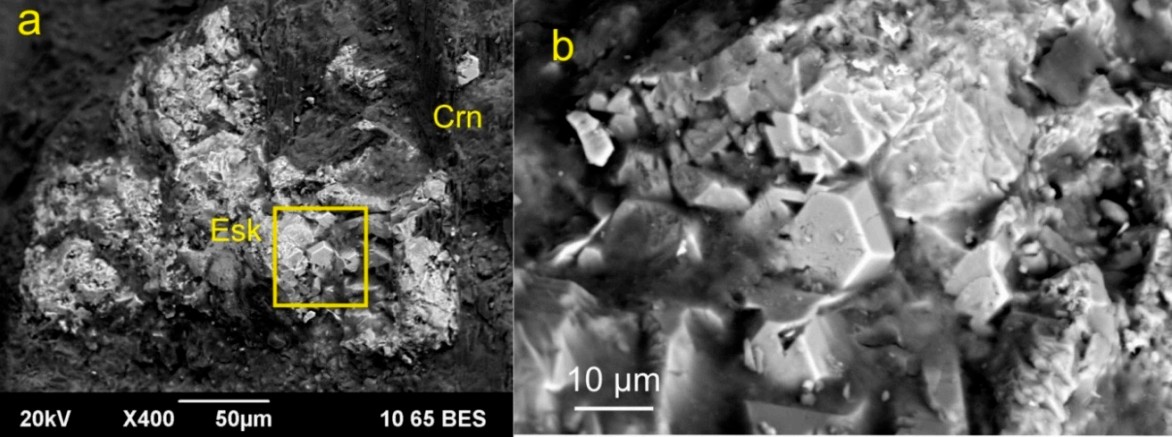

**Figure 3.** The aggregation of eskolaite crystals (Esk) on the surface of a ruby (Crn) crystal: (**a**) general view and (**b**) a close-up of the yellow fragment depicted in Figure 3a.

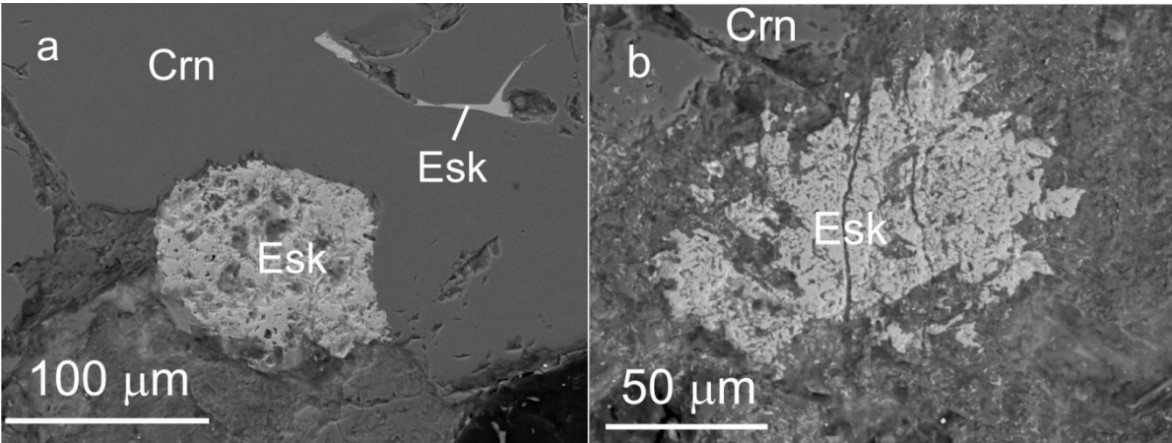

**Figure 4.** Back-scattered electron (BES) image of the eskolaite (Esk) (**a**) on type 1 ruby and (**b**) on type 3 ruby.

## 5. Discussion

### 5.1. PT-Conditions for the Formation of Ruby and Eskolaite Association in the Kuchinskoe Occurrence

At the peak of the regional metamorphism of carbonate rocks in the Kuchinskoe marble-hosted ruby occurrence, the temperature reached 660 °C (calcite-dolomite geothermobarometer [36]) at a pressure of 2.6–1.2 kbar [7]. The temperature of type 3 ruby formation is estimated as 600–550 °C, p = 2.2–1.9 kbar, and $p_{CO_2}$ = 1.4–0.4 kbar. PT conditions for the formation of type 1 ruby have not yet been evaluated. However, the temperature was probably in the range of 550–600 °C and the pressure could reach 3–4 kbar. These PT conditions for the formation of ruby-bearing marbles of the Kuchinskoe occurrence are in good agreement with the results of the studies of regional metamorphism in the Kochkar anticlinorium using mineral paragenesis (Garnet + Biotite + Plagioclase + Quartz ± Staurolite ± Sillimanite ± Cordierite): 570–640 °C, 3–5 kbar [28]; 590–620 °C, 2.5–3.0 kbar [29]; and 500–620 °C, 3.0–4.0 kbar (in inter-dome structures) and 620–700 °C (in domes) [30]. The pressure during the retrograde metamorphism is estimated by the depth of formation of the Kochkar anticlinorium, which is 4.0–4.5 km (about 1.5 kbar) [28]. At the Kuchinskoe occurrence, the eskolaite is only associated with high-chromium rubies, which possibly indicates the genetic relationship between these minerals. To estimate the temperature of eskolaite formation, a geothermometer was proposed based on the experimental data on the stability of corundum-eskolaite solid solution at various PT parameters [37]. The molar fractions of $Cr_2O_3$ eskolaite crystals ($X_{Cr2O3}$) on the rubies of prograde and retrograde stages, according to their crystal-chemical formulas, are 0.65–0.8 (see Table 2). Such $X_{Cr2O3}$ values for eskolaite, subject to their homogeneity, make it possible to estimate the formation temperature of this mineral by the corundum-eskolaite geothermometer as 700–850 °C Such a high temperature excludes the formation of eskolaite in karst conditions. At the same time, it is higher than the estimate of the temperatures of metamorphism in the Kochkar anticlinorium (500–670 °C). The mismatch in the temperature parameter of the ruby and eskolaite formation is still difficult to explain. This discrepancy may be due to the influence of oxygen fugacity, pH, or other factors.

### 5.2. Aluminum and Chromium Sources in the Formation of Ruby and Eskolaite in Marbles

1. The following geological data dispute the sedimentary origin of Al and Cr in ruby-containing marbles [1–5]:

(i) there is no evidence of sedimentary lamination in ruby-containing marbles;

(ii) in the Kochkar anticlinorium, the length of a strip of carbonate rocks is measured in the tens of kilometers, with a width of up to 2.7 km. The marbles of this strip contain an admixture of Al and Cr and are in the same zone of metamorphism, but ruby mineralization occurs very locally;

(iii) the length of the areas of ruby mineralization in marbles rarely exceeds 250–300 m along the flow cleavage and their width does not exceed 80 m. The vertical scale of mineralization has not been established, but is the greatest in small bodies. Thus, ruby mineralization forms flattened lenticular columnar bodies and this form of ruby-bearing marbles contradicts the concept of stratified sediments;

(iv) the formation of rubies in marble occurred both at the prograde and retrograde metamorphism, and was not one-act (see geological setting): 2–3 types of ruby can be seen in one marble sample, which requires the changes of conditions during their formation.

But we did not exclude the participation of Al and Cr of sedimentary origin in the formation of rubies, provided that they were dissolved by metamorphogenic fluids.

2. Desilication of sedimentary terrigenous-carbonate rocks during regional alkaline metasomatism [6] could take place. But there are no signs of marble desilication in the Kuchinskoe occurrence. In addition, the formation of rubies occurred during the prograde and retrograde metamorphism.

3. The introduction of deep fluids into Al marbles by the gas phase during alkaline magmatism [9] or decompression during deep tectonic erosion of the Earth's crust [9] cannot be applied for the Kuchinskoe marble-hosted ruby occurrence since there is no alkaline magmatism and deep tectonic erosion here.

4. The redistribution of Al and Cr during metamorphism of sedimentary limestones with evaporate lenses [10,11] does not contradict facts (i–iii), but does not explain fact (iv). Mg-metasomatism in the marbles of the Kochkar anticlinorium manifested twice: (1) through organogenic limestones at an early prograde metamorphism with the formation of fine-grained dolomites and (2) through Mg-calcite and calcite marbles at an early retrograde metamorphism, with the formation of medium-grained dolomite-calcite marbles. Mg metasomatism of the prograde metamorphism is probably caused by the destruction of evaporites where metasomatic dolomites of this stage are enriched in Al, Cr, V, Cu, Co, and REE. By contrast, two-carbonate marbles are depleted in these elements.

5. The introduction of Al and Cr by metamorphogenic fluids during rock granitization [7,8] does not contradict the facts (i–iv). By "granitization" we mean a high temperature metasomatic process that brings the initial rock closer to granite in chemical composition through the introduction of $SiO_2$, Na, K, $H_2O$, and the removal of Mg, Fe, Ca and other femic components. As a possible cause of granitization, we considered the deformation of the continental crust during Late Paleozoic collision [30,34,38], the times of tectonic deformations, dome formation, and metamorphism in the Kochkar anticlinorium coincide [28–30]. They culminated in the introduction of anatectic granites and pegmatites. The rocks of the dome structures are enriched with fluorophilic elements (Be, Li, Sn, Ta, Nb, and others) and the rocks of the inter-dome structures are rich in Fe, Mg, Ca, Ti, and others. The substrate was volcanic-sedimentary rocks, distributed around the Kochkar anticlinorium [27].

6. The solution for the problem of the Cr and Al sources for the formation of red corundum in the hyperbasite Rai-Iz massif (the Polar Urals) was recently considered [39]. The authors [39] believe that the vein was the product of the interaction between a subduction-zone-derived fluid and mantle wedge peridotite. The Cr content of the red corundum was potentially derived from accessory chromian spinel in the peridotite. Popov et al. [40] considered that the chromium-bearing spinels of the Bazhenovsky ophiolite complex are the main chromium sources for the coloring of emeralds and alexandrites of the Mariinsky deposit (Middle Urals), located 15 km west. We did not comment on these statements, since only ruby-containing marbles were considered in this article.

7. In Mg-calcite (ruby-bearing) marble, the content of Al and Cr is low at 0.11–0.13 wt % $Al_2O_3$ and 0.00096–0.0014 wt % $Cr_2O_3$ [8]. Marbleized bituminous organogenic limestones in the marginal parts of the anticlinorium (about 5 km from the Kuchinskoe occurrence) contain 3-times greater the amount of Al and Cr. The primary chemical composition of limestones changed significantly during the prograde and initial retrograde metamorphism [8,32,34]. Evaporitic lenses, if present in limestone (as suggested in [11,12]), could also undergo these processes.

We believe that ruby and eskolaite in the Kuchinskoe occurrence were formed by the hydrothermal-metasomatic process. Eskolaite was observed only on the surface of ruby crystals or in

their near-surface region. This is probably due to the formation of eskolaite only at the final stage of the ruby crystal growth. Chromium zonal distribution in ruby crystals was not observed. Corundum enrichment (or depletion) in chromium in the contact zone with eskolaite was sometimes present, but is not a rule. The reason for this has not yet been established. Probably, the chromium content in corundum is limited by crystallization conditions rather than by chromium activity. This corresponds to the observed facts: the spinel and ruby have maximum Cr contents at the prograde stage, minimal Cr concentration at transitional stage, and again enriched in Cr at the retrograde stage. Sazonov [41] noted that $Al^{3+}$ and $Cr^{3+}$ have amphoteric properties and migrate under the similar conditions in hydrothermal metasomatic processes. In this case, we can assume that Al and Cr were introduced into the marble by high-temperature fluid. The metamorphism of sedimentary carbonate rocks in the Kochkar anticlinorium was not isochemical [7,8] and Cr was a component of many accessory minerals in marbles. Consequently, eskolaite could not form under the conditions of isochemical metamorphism. Due to the evolution of metamorphic conditions, the introduction of Al into the system ceased, but Cr still entered, forming crystalline crusts on the ruby.

## 6. Conclusions

The identification of eskolaite in association with rubies in the marbles of the Kuchinskoe occurrence is a new argument in favor of the introduction of Al and Cr into the mineral formation zone. Here, eskolaite was associated only with high-chromium rubies of the prograde and retrograde metamorphism, which confirms the migration of these elements in similar conditions. The formation of eskolaite occurs at the final stage of the ruby crystal growth and later. The eskolaite of prograde stage contains an admixture of titanium, which is explained by its low activity and inability to form its own minerals. The occurrence of rutile and sphene at the retrograde stage explains the low titanium content in the recent eskolaite. The parameters and morphology of the mineralized zones also indicate the introduction of Al and Cr by metamorphogenic fluids.

**Author Contributions:** Conceptualization, investigation and writing—original draft preparation, A.K.; writing-review and "Results" and "Discussion" section editing, supervision, V.M.; Powder X-Ray diffraction analysis and structure solution, S.S.; SEM and EMPA analyses, I.G.; writing—review, "Materials and Methods" section editing and translation, D.K. All authors have read and agreed to the published version of the manuscript.

**Funding:** The work was performed within the framework of topic No. AAAA-A18-118052590028-9 of the IGG UB RAS state assignment; SEM and EMPA analyses were carried out at the Geoanalitik UB RAS Collective Use Center within the framework of topic No. AAAA-A18-118053090045-8 of the IGG UB RAS state assignment.

**Acknowledgments:** The Editor of *Minerals* and Guest Editors are thanked for organizing the Special Issue on "Mineralogy and Geochemistry of Ruby and Allied Assemblages" and extending an invitation to submit a contribution for consideration. This particularly includes Frederick Lin Sutherland and Khin Zaw. We thank the Reviewers for the important comments and constructive suggestions, which helped us to improve the quality of the manuscript.

**Conflicts of Interest:** The authors declare no conflict of interest. The funders had no role in the design of the study; in the collection, analyses, or interpretation of data; in the writing of the manuscript, or in the decision to publish the results.

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
