# Peer review of "The First Find of Cr2O3 Eskolaite Associated with Marble-Hosted Ruby in the Southern Urals and the Problem of Al and Cr Sources"

_minerals, doi:10.3390/min10020101_

Round 1

Reviewer 1 Report

Review of the article “The first find of Cr2O3 eskolaite associated with marble-hosted ruby in Southern Urals and the problem of Al and Cr sources” by Kissin A. et al.

The article by Kissin et al. presents the discovery and geochemical characterization the rare mineral eskolaite found intergrowing with the ruby crystals in the Kuchinskoe deposit from Russian Southern Urals. This mineral was assumed by authors to be the evidence for Cr and Al introduction in the corundum formation zone at this location. I suppose the short length of the article is in the acceptable scope of research; meanwhile, there are several ways how the manuscript could be strengthened. Firstly, the interpretation for some results is novel, however, out of the up-to-date knowledge in corundum mineralogy and geochemistry. Secondary, I have great concern regarding the origin of studied ruby crystals and their further methodological description. Thus, I recommend the acceptance of the article after a MAJOR REVISION assuming the authors have taken into consideration the following recommendations:

MAJOR COMMENTS:

The authors stated the metamorphic origin for studied corundum association. However, I had a chance seeing previously the ruby samples embedded in the marble matrix from Kuchinskoe location reaching a maximum of a few mm in length and being mostly anhedral by morphology. Meanwhile, there is also a secondary location where the larger material was found as well. Therefore, it is not clear from the text which location type was described by authors in the manuscript. If the location is secondary assuming from the ruby size and morphology in Fig. 2b, then, the proof for metamorphic origin for ruby and eskolaite should be clearly provided in the text including the photos and dimension for all studied ruby samples, ruby geochemistry using the discrimination diagrams (Giuliani et al. 2014, Sutherland et al., 2019, etc.), their solid and/or fluid inclusion scenes, detailed mineralogy, and/or oxygen and radiogenic isotopic composition for both corundum and eskolaite, etc. Currently, there is no evidence in the text confirming the metamorphic origin in co-precipitated mineral pair.

Giuliani G., Caumon G., Rakotosamizanany S., Ohnenstetter D., Rakototondrazafy M. (2014). Revue de Gemmologie 188, 14–22

Sutherland F.L., Zaw K., Meffre S., Thompson J.; Goemann K.; Thu K., Nu T.T., Zin M.M., Harris S.J. (2019). Minerals 9, 28.

If the proof on their metamorphic origin would be provided, then, the sampling strategy should be better stated in the text. Currently, it is hard to follow which rock type was the source for studied minerals. What are rubies of type 1, type 2, type 3? Their mineralogical description needs to be provided in the text. How the metamorphic degree and stage were defined for them? Which method was used in this regard? Moreover, the methodological sequence should be better described in the text, e.g., currently, it is not clear how many eskolaite samples were analyzed by EMPA and further controlled by XRD? I could find only 2 XRD measurements provided in Table 1 comparing to 14 EMPA analyses in Table 2, and even though, it is not clear which samples from Table 1 are which in Table 2. For instance, the extremally low Cr2O3 concentrations for the second eskolaite measurement in Table 2 likely outnumbering Al2O3 in atomic numbers is not likely the case of eskolaite. This may be a result on contamination between corundum and eskolaite or targeting the corundum with extreme Cr concentration during EMPA measurements reported previously for Israel placer deposit, however, formed likely in the super-reduced conditions (Gain et al. 2019), which is not the case for studied Kuchinskoe Therefore, the proof for each eskolaite chemical analysis in Table 2 should be better stated including recalculation on atomic numbers with appropriate Raman or XRD data at least in Supplementary materials. Finally, the exact mechanism for eskolaite and corundum co-precipitation should be better described in the text. As it is seen from the manuscript, eskolaite formation occurred on preexisting corundum crystal with their further co-precipitation. However, following this scenario, I would expect the finding at least the corundum solid inclusion within the eskolaite host which is likely not the case. Otherwise, the only way for it near terminate surface precipitation on preexisting corundum crystals is its epitaxial growth along the specific ruby face. If this is the case for studied samples, then, the detailed crystallographic description in regard to intergrowing faces for both eskolaite and corundum should be better stated in the text.

Gain S. E. M., Griffin W. L., Saunders M., Toledo V. (2019). Microscopy and Microanalysis, 25(S2), 2484-2485.

Concluding, the article is lack on citing the latest publications on corundum and emerald geochemistry including the alternative opinion in regard the source of chromophores, i.e. Cr-bearing spinel inclusions acting as Cr, Fe, and, in some cases, V source for hosts coming from nearby ophiolitic complexes in the Middle and Polar Urals (Meng et al. 2018, Popov et al. 2018). Besides, the article would benefit by the citation on some of the publications regarding the other corundum locations even magmatic in origin, however, also located in South Urals with minor Cr concentrations, but yet without detected possible source of chromophores (Sorokina et al., 2017, Filina et al. 2017, Sorokina et al. 2019). As well as the comparison the Kuchinskoe occurrence with those known marble-hosted ruby deposits would fall the manuscript in a global context being more valuable for wider geological and mineralogical communities, e.g. those briefly described within Alpine-Himalayan belt by Giuliani et al. 2014 and references therein.

Meng F., Shmelev V.R., Kulikova K.V., Ren Yu. (2018) Lithos (320–321), 302-314

Popov M.P., Sorokina E.S., Kononkova N.N., Nikolaev A.G., Karampelas S. (2019) Doklady Earth Sciences 486(2), 630 – 633.

Sorokina E.S., Karampelas S., Nishanbaev T.P., Nikandrov S.N., Semiannikov B.S. (2017). Canadian Mineralogist 55, 823–843

Filina M.I., Sorokina E.S., Botcharnikov R., Karampelas S., Rassomakhin M.A., Kononkova N.N., Nikolaev A.G., Berndt J., Hofmeister W. (2019). Minerals 9, 234.

Sorokina E.S., Rassomakhin M.A., Nikandrov S.N., Karampelas S., Kononkova N.N., Nikolaev A.G., Anosova M.O., Somsikova A.V., Kostitsyn Y.A., Kotlyarov V.A. (2019). Minerals 9, 36.

Giuliani G., Ohnenstetter D., Fallick A.E., Groat L., Fagan A.G. (2014). In: Groat, L.A. (Ed), Geology of Gem Deposits, 2nd ed. Mineralogical Association of Canada, Tucson, pp. 29–112.

MINOR COMMENTS:

Introduction: The scientific aim of ruby origin research should better be stated in the text.

L 31 What about the other trace elements in ruby, e.g. Fe, Ti, Mg, Ga, and Si? Is their source clear in marble-hosted locations?

L 38 – 39. The 7th theory was proposed firstly by Spiridonov (1998) for numerous locations in Pamir and Uralian folded areas (Spiridonov E.M., 1998. Gemstone deposits of the former Soviet Union. J. Gemmol. 26, 111–124).

L 55. Oxygen fugacity is a function of temperature in logarithmic values.

L 78 – 86 The appropriate citations need to be provided in the text

Figure 1a. There is not a correct signing in the map: it should be just the Urals or the Ural Mountains, not a combination of both. Please carefully check of the signs in the map b. I even not sure about Zauralovskiy synclinorium. Should it be Zauralsky?

L 101 – 104. Please see the major comments regarding the sampling strategy.

L 110 Where are these samples in Table 2?

L 112 – 113. Please provide the crystals used in WDS and conditions of measurements such as accelerating voltage, beam current, counting time, beam size along with the reference materials used for calibration/measurements, overlap corrections, 2 sigma uncertainties, and detection limits for each measured element.

L 115. How many samples were studied and where are they in Table 2?

L 116 BSE images or images in secondary electrons? It should be stated in the text.

L 118. Again, please provide the conditions of measurements.

L 119. How many samples were studied by XRD? Where are they in Table 2?

L 121 What is the step of measurements? Which database was used for interpretation?

L 123 How many of these grains were analyzed and how many identifications were obtained? What are the concentrations of trace elements? It should be provided in the text.

L 128 In which samples from Table 2?

L 128 Electron micro-probe analyses?

L 132 Something is wrong with punctuation at this figure caption,

L 135 Fig. 3 a, b – are these secondary electron images? Please provide the type of image in the figure caption. Are these samples from Fig. 2?

L 139 The analysis with 54.27 wt.% of Al2O3 is not likely eskolaite and needs to be controlled.

Table 1: What does it mean 128.312 in hkl column?

Table 2: The second measurement is not likely eskolaite. The 5th, 13th, and 20th lines with Al2O3 values of 95.95 wt.%, 95.79 wt.%, and 93.95 wt.% are too low for corundum. Where is the total sum for these ruby measurements? Please provide these measurements in the separate lines and after, the average.

The Ga trace element used for corundum origin determination was not provided in table 2.

Discussion: this section should be almost completely rewritten based on the major comments described above.

L 157 – 160. How were these PT conditions estimated? By which method? It should be clear in the text.

L 161 – 162 What does it mean the high chromium? What is the dimension for low and high? The exact Cr values should be stated in the text.

L 162 – 163. I didn’t understand this sentence. Please rephrase. Why the exogenic eskolaite formation is unlikely? By which reason? Please provide an explanation in the text.

L 164 – 185 This text is hard to follow and needs to be comprised in the comparison table.

L 165 There is too much Al2O3 of 54.27 wt.% within measured eskolaite.

L 167 There are only 6 analyses in Table 2.

L 169 – 170. I didn’t understand this sentence. Equal amount to what? Please rephrase.

L 172 – 174. The maximum Cr content in Kuchinskoe ruby samples is 2.97 wt.%. Then what brings it close to those found in Udachnaya kimberlite? Please rephrase the sentence. This even hard to follow. Please provide the comparison table with those eskolaites detected in the other locations.

L 179 – 180 In which units the measured Al2O3 and Fe2O3 contents, wt.%? Please state it in the text and provide the equivalent values in wt.% for V2O3.

L 181 – 183 I didn’t understand this sentence. Please rephrase it.

L 187 – 190 Why the Al and Cr cannot be precipitated in bauxite-like protolith and further metamorphosed as in the other marble-hosted occurrences? All the theories provided in the introduction section should be clearly discussed in the text.

L 191 – 193. Please see the major comment 2. This should be deeper discussed in the text.

L 195 – 196 I didn’t understand this sentence. Please rephrase.

L 199 – 213 These statements should be confirmed not only by amphoteric properties for both Al and Cr, marble cleavages or so but the analytical measurements, e.g. oxygen isotopic composition in corundum and eskolaite, etc. Currently, it looks just like the authors’ fantasy. I would omit these unnecessary speculations.

L 207 – 208 Where are the units for the PT scale? What do these arrows mean? Please state it in the figure caption.

Reviewer 2 Report

Kissin et al have submitted an interesting manuscript about a rare association of eskolaite with a marble-hosted ruby occurrence in the Southern Urals and the source of the Al and Cr that formed the ruby and eskolaite. They have found three stages of ruby mineralization (presumably together with associated eskolaite mineralization) beginning with a prograde stage (type 1) a transitional stage (type 2) and a retrograde or hydrothermal stage (type 3). The authors have carefully analyzed the ruby and associated eskolaite and have found clear differences in the composition of these minerals that they interpreted as due to contrasting prograde and retrograde environments. This is an interesting argument, however evidence in support of their conclusions are poorly presented and result in considerable and unnecessary confusion. The essence of their interpretation is a schematic diagram (Figure 5) that shows an inverse relationship between Cr content and PT (pressure and temperature) plotted against time. This interpretation is quite misleading and should be completely revised as pointed out in my line by line list of suggestions. It is plausible that the source of the Al and Cr was hydrothermal activity associated with granitic dome structures although very little evidence is provided. Is the ruby and eskolaite found in hydrothermal veins related to the domes? Is there any other hydrothermal mineralization. Is there evidence of ruby mineralization found in contact metamorphic zones near the granitic domes such as a skarn assemblage? Without such information we are left with random and unsupported statements such as “The mineralization is controlled by cleavage cracks, and not by rock layering.” This may be an important observation but needs much more explaniation. Typically any cracking is evidence of retrograde metamorphic activity while prograde rock is typically uncracked (not to be confused with unveined). And if the source of Al to form ruby mineralization is not associated with rock layering, at least some evidence for an absence of such correlation should be presented. In short, the manuscript should be revised so that clear evidence is inserted before all interpretations. Otherwise it is just speculation. Line by line suggestions follow:

Line 17. The word “and” should follow the word “hexagonal”.

Line 17. A sentence such as “Both prograde and retrograde ruby mineralization has been identified” should be inserted here.

Line 24. A phrase such as “… granitic dome sourced metasomatic and hydrothermal…” should follow the word “of”. Otherwise there is no clue as to what you are talking about.

Line 26. They are all crustal rocks and no reference to the granitization process is discussed in the text. I suggest deleting the whole sentence. It only adds confussion.

Line 36. The word “granitization” was originally proposed as an almost magical transformation of rock into granite without any basis in fact and has been rarely used since the 1940s. It is typically used in current literature as a substitute for “in-situ anataxis” but it is always unclear and should not be used without an explanation. 

Line 37. Decompression is not a source of Al or Cr but decompression-melting can result in granite domes that could be a source of hydrothermal Al and Cr.

Line 57. This sentience is confusing. It could be shortened to “Eskolaite was also found in kimberlites.”

Lines 89 to 93. The types should be clearly specified. Insert (type 1) into line 89, (type 2) into line 90 and (type 3) into line 90.

Line 133. On the basis of figures 2 and 3 it will appear to many petrologists that the ruby was supersatursted in Cr and simply underwent eskolaite exsolution when it was cooled to the solvus described by Chatterjee et al.

Line 149. The grains are badly pitted. This can result in considerable contamination during probe analysis. You should indicate in the methods section how you made sure your probe data is accurate despite these problems.

Lines 154 to 160 and Figure 11. If the composition of ruby formation is controlled by PT conditions then the composition of ruby during prograde metamorphism should keep changing until peak PT conditions are met. Your type 1 (prograde) composition should therefore reflect peak PT conditions (the highest values). Type 1 must, therefore, reflect the peak of your PT curve by definition. Type 2 or the transition PT conditions must plot on the beginning of the right-handed downslope of the curve and retrograde must plot further down the slope toward increasing time. Figure 11 simply isn’t consistent with your type 1 –type 3 interpretation.

Line 183. Chatterjee et al have demonstrated that they have an “excellent geothermometer” based on corundum-eskolaite solid solution. Did you try to use it? I would think it would supply valuable information.

Line 195. Some of the eskolaite in figure 4a looks like exsolution lamellae.

Line 198. How can you be sure that type 2 actually represent a transition stage? Again - What is the evidence?

Line 184. Do you actually see any rutile in type 2 and 3?

Line 211. What do you mean by “cleavage cracks”? Describe them.

Reviewer 3 Report

Mineralisation in the Kuchinskoe anticline is relatively well studied, however the occurrence of eskolaite in this area is not. As such, a description of this phase certainly does contribute to the greater understanding of the area. 

The authors state several times "V2O5 Karelianite", however as I mentioned in the paper, Karelianite is V2O3 and shcherbinaite in V2O5. I'm am uncertain which phase the authors mean and based on the lack of characterising evidence on this phase, I cannot accurately guess.

In its current form, the "material and methods" chapter is significantly lacking details with regards to analytical parameters.

The manuscript could benefit from a comparative geochemical plot illustrating the eskolaite varieties found and comparing them to global analogues - rather than simply mentioning their chemistry.

The data obtained by the authors is good but needs to be ordered and written in a more scientific manner. I strongly advise they resubmit after some major reworking.

Round 2

Reviewer 1 Report

Review of resubmitted article “The first find of Cr2O3 eskolaite associated with marble-hosted ruby in Southern Urals and the problem of Al and Cr sources” by Kissin et al.

The resubmitted manuscript is significantly better than the previous version and presents a deeper discussion regarding the problem of Al and Cr sources in Kuchinskoe ruby deposits from South Urals. However, numerous other claims were still not fixed in the current text. For instance:

The manuscript still remains highly debatable regarding the ruby origin in marbles and karst occurrences. Oxygen isotopy is a good argument; however, I didn’t find any values for these two deposits in Vysotskiy et al. paper. Therefore, the comparison table including ruby trace element chemistry (Cr, Ti, Fe, Ga, V, and Mg), solid and fluid inclusions, mineral paragenesis, oxygen isotopes is still required for both occurrences, at least in the supplementary material. Besides, the exact mechanism explaining the eskolaite precipitation on pre-existing ruby crystals is still absent in the text. Please provide this information. Moreover, as I see from the current revised version, the article presents mostly a theory on Cr source rather than Al origin. If this is the case, please revise the text in accordance. Otherwise, the authors need providing the formation on process explaining how to obtain more Al-enriched corundum from Al-depleted eskolaite mineral. The article is still lack on citing the publications providing the alternative opinion on Cr source in emerald and ruby deposits from Middle and Polar Urals. If to consider the references in the introduction section, the authors create an opinion for readers that eskolaite is only an exclusive Cr source for these deposits, which is not true. Therefore, I suggest including the references on the other publications providing the alternative opinion on Cr sources at these two deposits, e.g. Meng et al. and Popov et al. or others. The chapter by Giuliani et al. (2014) on “The geology and genesis of gem corundum deposits” doesn’t cite these articles appeared much later after the publication of this Special issue of Canadian Mineralogical Society.

Meng F., Shmelev V.R., Kulikova K.V., Ren Yu. (2018) Lithos (320–321), 302-314

Popov M.P., Sorokina E.S., Kononkova N.N., Nikolaev A.G., Karampelas S. (2019) Doklady Earth Sciences 486(2), 630 – 633.

Summarising all the above, I would still suggest accepting the manuscript after MAJOR REVISION assuming that authors taking into consideration the abovementioned suggestions in the new version of the manuscript.

MINOR COMMENTS:

L 167 Why only one eskolaite sample was analyzed by XRD? Where is the proof that all the other samples in Table 2 are eskolaite?

L 174 – 175. Why only Ti to V correction was applied? How about Cr and V overlapping?

L 177 What does it mean 10/5, 30/15, 60/30? Please revise.

L 211 Please provide the confirmation that anomalous measurement of Cr2O3 (up to 40.8wt.%) belongs to corundum? Any XRD of Raman analyses?

Table2 Should n/d be the values below the detection limit (bdl)? Please revise.

Some of the values in Table 2 at the detection limit. Please revise them taking into account the uncertainties and two detection limits for all elements.

Besides, I am not sure that Table 2 was completely revised since I still don’t understand where the eskolaite sample from Table 1 in Table 2.

L 236 Which mineral paragenesis? Please state this in the text.

L 259 – 262. I didn’t understand this sentence. Please rephrase.

L 276 What does it mean? Please provide a deeper discussion.

L 277 – 278 I didn’t understand this sentence. Please rephrase.

L 287 – 288 I didn’t understand why the fact iiii contradicts the theory on occurring the lenses of evaporites within marble layers.

L 300 Please use the accepted dimensions of numbers.

Author Response

Please see attachment/

Reviewer 2 Report

The revised version is an improvement over the original draft.  The English is still in need of some improvement. At least the abstract should be good English so line 11 should read "hexagonal and tabular" and Line 27 should read "introduction of Al and Cr into the …" Otherwise the manuscript looks OK. 

Reviewer 3 Report

The authors provide evidence for the first find of Cr2O3 eskolaite associated with marble-hosted ruby in the southern Urals. An in-depth characterisation has been conducted on several samples that satisfy the phase identification and geochemical quality. Methods and procedures have been outlined adequately and the result presented in a logical manner. The authors have exercised due diligence in their analysis of previous work, providing a suitable comparison. The authors have considered both the hydrothermal and tectonic facets of phase formation in their analysis in order to justify their conclusion.

I am satisfied that the authors have addressed the major concerns outlined previously and that the manuscript (with a few minor corrections) is suitable for publication. 
